

# Genetic diversity increases with depth in red gorgonian populations of the Mediterranean Sea and the Atlantic Ocean

Joanna Pilczynska[1,2], Silvia Cocito[3], Joana Boavida[4,5], Ester A. Serrão[4],
Jorge Assis[4], Eliza Fragkopoulou[4] and Henrique Queiroga[1]

[1] Departamento de Biologia and CESAM—Centro de Estudos do Ambiente e do Mar, Universidade de Aveiro, Aveiro, Portugal
[2] Department of Earth and Environmental Sciences, University of Pavia, Pavia, Italy
[3] Italian Agency for New Technologies, Energy and Sustainable Economic Development—ENEA, Marine Environment Research Centre, La Spezia, Italy
[4] CCMAR—Centro de Ciências do Mar, Universidade do Algarve, Faro, Portugal
[5] Aix Marseille Université, CNRS/INSU, Université de Toulon, IRD, Mediterranean Institute of Oceanography (MIO) UM 110, Marseille, France

Corresponding author
Eliza Fragkopoulou,
efragkopoulou@ualg.pt

## ABSTRACT

**Background**. In the ocean, the variability of environmental conditions found along depth gradients exposes populations to contrasting levels of perturbation, which can be reflected in the overall patterns of species genetic diversity. At shallow sites, resource availability may structure large, persistent and well-connected populations with higher levels of diversity. In contrast, the more extreme conditions, such as thermal stress during heat waves, can lead to population bottlenecks and genetic erosion, inverting the natural expectation. Here we examine how genetic diversity varies along depth for a long-lived, important ecosystem-structuring species, the red gorgonian, *Paramuricea clavata*.

**Methods**. We used five polymorphic microsatellite markers to infer differences in genetic diversity and differentiation, and to detect bottleneck signs between shallow and deeper populations across the Atlantic Ocean and the Mediterranean Sea. We further explored the potential relationship between depth and environmental gradients (temperature, ocean currents, productivity and slope) on the observed patterns of diversity by means of generalized linear mixed models.

**Results**. An overall pattern of higher genetic diversity was found in the deeper sites of the Atlantic Ocean and the Mediterranean Sea. This pattern was largely explained by bottom temperatures, with a linear pattern of decreasing genetic diversity with increasing thermal stress. Genetic differentiation patterns showed higher gene flow within sites (i.e., shallow vs. deeper populations) than between sites. Recent genetic bottlenecks were found in two populations of shallow depths.

**Discussion**. Our results highlight the role of deep refugial populations safeguarding higher and unique genetic diversity for marine structuring species. Theoretical regression modelling demonstrated how thermal stress alone may reduce population sizes and diversity levels of shallow water populations. In fact, the examination of time series on a daily basis showed the upper water masses repeatedly reaching lethal temperatures for *P. clavata*. Differentiation patterns showed that the deep richer populations are

isolated. Gene flow was also inferred across different depths; however, not in sufficient levels to offset the detrimental effects of surface environmental conditions on genetic diversity. The identification of deep isolated areas with high conservation value for the red gorgonian represents an important step in the face of ongoing and future climate changes.

## INTRODUCTION

Extreme environmental conditions may change the distribution of intra-specific biodiversity (*Provan & Bennett, 2008*). However, responses may differ significantly between environments and ecological groups. The trends in genetic diversity of mountain plants and vertebrates along altitudinal gradients are a well-known example, varying from decreased diversity with altitude due to drift and bottlenecks during vertical range expansion, to increased diversity associated with selective pressures at higher altitudes (*Giordano, Ridenhour & Storfer, 2007*; *Ohsawa & Ide, 2008*).

In the marine environment, environmental gradients are known to affect the genetic diversity levels of the populations (e.g., *Costantini et al., 2011*; *Costantini et al., 2016*; *Johannesson & André, 2006*). Particularly, the exposure to limiting niche conditions may reduce population sizes, leading to genetic erosion through bottlenecks and drift (*Eckert, Samis & Lougheed, 2008*). Conversely, where conditions are stable for long time, populations may persist and retain ancient genetic diversity (i.e., climatic refugia; *Maggs et al., 2008*; *Provan & Bennett, 2008*). Even small populations of species with reduced dispersal potential may harbour distinct ancient genetic diversity if persisting in refugial areas (e.g., *Diekmann & Serrão, 2012*; *Assis et al., 2016*).

In the coastal zone, depth gradients are associated with increased environmental variability due to the stratification of wind-induced turbulence, light attenuation, nutrient availability, sedimentation and the presence of thermoclines and haloclines (*Garrabou, Ballesteros & Zabala, 2002*; *Assis et al., 2017*). Together, they underpin well-structured gradients of species occurrences, abundances and genetic diversity. While extreme conditions experienced in shallow waters can be detrimental to genetic diversity, deep waters are relatively more stable and can provide genetic refugia (e.g., *Smith et al., 2014*; *Assis et al., 2016*). The hypothesis of deep populations harbouring higher and unique genetic diversity is of great conservational, biogeographical and evolutionary relevance. The loss of such genetically rich populations poses disproportionate risks for the species as a whole, considering the loss of adaptive variation for selection (*Hampe & Petit, 2005*). While there is increased evidence of deep populations not being directly affected by events of extreme environmental conditions (e.g., *Cerrano et al., 2005*; *Linares et al., 2005*; *Smith et al., 2014*), these are not immune to disturbances (*Bavestrello et al., 2014*; *Frade et al., 2018*). Furthermore, the hypothesis that deeper reefs may be more fecund (*Holstein et al.,*

*2015*), serving as a source of recruits for the recovery of shallower reefs is controversial between species (*Bongaerts et al., 2010*; *Bongaerts et al., 2017*), but also within the same species (*Mokhtar-Jamaï et al., 2011*; *Van Oppen et al., 2011*; *Pilczynska et al., 2016*).

The red gorgonian *Paramuricea clavata* (Risso) lives on shadowed rocky substrates down to 120 m in the Mediterranean Sea (*Salomidi et al., 2009*; *Bo et al., 2012*) and to 100 m in the Atlantic Ocean (*Boavida et al., 2016a*). Shallow populations in the Mediterranean have recently suffered massive mortality events caused by thermal stress (*Perez et al., 2000*; *Romano et al., 2000*; *Garrabou et al., 2009*). Damage intensity has been reported to decrease with depth, with communities dwelling below the thermocline of 25 to 30 m being less affected (*Cerrano et al., 2005*; *Linares et al., 2005*). In the Atlantic Ocean, temperature-driven mortality events are presumed to be less frequent as water masses are usually mixed due to summer upwelling (*Relvas et al., 2007*). However, there are sporadic events of upwelling relaxation, with raising temperatures persisting for several days, responsible for the mortality of shallow water gastropod populations (*Lima et al., 2006*) and marine forests (*Araújo et al., 2016*), which eventually may reach levels that could become limiting for the red gorgonian. Here we investigate changes in the genetic diversity levels of *P. clavata* across populations in different depth ranges from the Mediterranean Sea and the Atlantic Ocean. Considering the different potential effect of temperature extremes between different depths, we hypothesize that shallow populations have lower genetic diversity when compared to deeper ones. We further explore the observed patterns of diversity using generalized linear mixed models with environmental information (temperature, current velocity, primary productivity and slope).

## MATERIALS & METHODS

We sampled two populations at different depths (shallow vs. deeper) in three sites of the Atlantic Ocean (Portuguese coast) and two sites of the Mediterranean Sea (Italy coast; Fig. 1; Table S1) by means of SCUBA diving. Atlantic samples were located between 12 m and 60 m depth, while those in the Mediterranean were between 20 m and 30 m. At each population, 10 cm apical branches were collected haphazardly from well-separated colonies to avoid clones, as the species can have asexual reproduction, though negligible (*Coma, Zabala & Gili, 1995*; *Pilczynska et al., 2017*). The branch tip from each colony was stored individually in plastic tube underwater and, after transportation to the lab, the samples were preserved in 96% ethanol until DNA extraction. Sampling permissions for the Mediterranean sites were authorised by Cinque Terre Marine Protected Area (*Pilczynska et al., 2016*). For the Atlantic sites, permission was authorized by the Institution for Conservation of Nature and Forest (ICNF, Portuguese governmental body responsible for the management of Protected Areas).

Genomic DNA was extracted with a CTAB protocol (*Winnepenninckx, Backeljau & De Wachter, 1993*), using proteinase K, with purification by standard chloroform:isoamyl alcohol (24:1) followed by DNA precipitation. Samples were genotyped at five microsatellite loci: Parcla 09, Parcla 10, Parcla 12, Parcla 14, Parcla 17 (*Mokhtar-Jamaï et al., 2011*). PCR conditions were as described in *Mokhtar-Jamaï et al. (2011)* with minor modifications (1.5

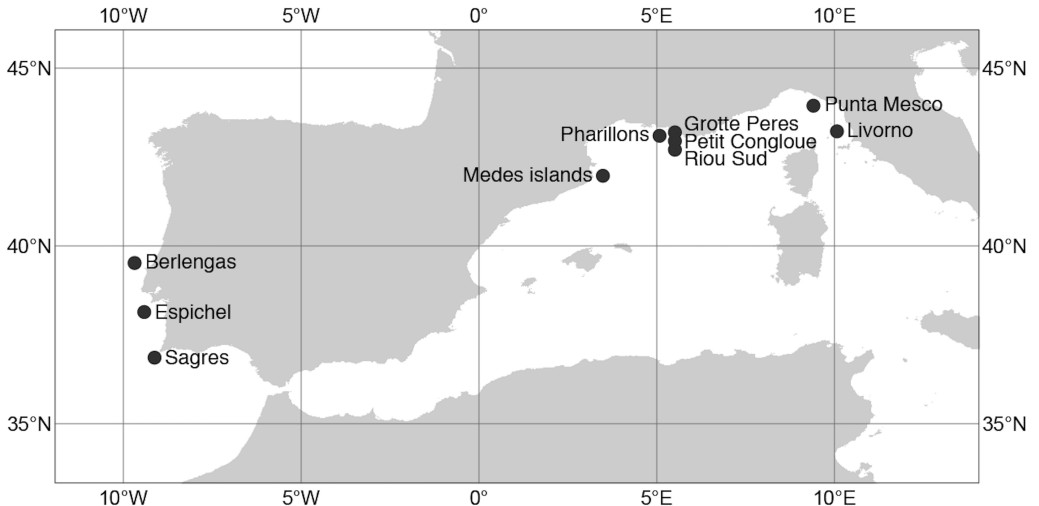

**Figure 1** **Sampling sites of the red gorgonian (*Paramuricea clavata*) populations in the Atlantic Ocean and the Mediterranean Sea.** The exact coordinates of each sampling site are available in the supplementary information (S3). Map source "©OpenStreetMap contributors", available under the Open Database License. This figure is published under CC BY SA: https://www.openstreetmap.org/copyright.

mM MgCl2, cycle: 95 °C 3 min, 94 °C 20 s, 45 °C 20 s, 72 °C 20 s for 40 cycles, final extension 72 °C 10 min). PCR products were analysed on an ABI 3730XL Genetic Analyser using an internal size standard (GeneScan 600 LIZ; Applied Biosystems). STRand version 2.2.30 was used to score alleles (*Locke, Baack & Toonen, 2000*) and the R package MsatAllel_1.02 (*Alberto, 2009*) allowed to visualise, track and reanalyse putative scoring errors.

Additional genetic data for the same microsatellite loci were compiled from *Mokhtar-Jamaï et al. (2011)*. Extra sites were chosen when comprising two populations at different depths (shallow vs. deeper). The new data (five sites) were located in the Spanish and French Mediterranean coasts, between 15 m and 40 m depth (Fig. 1; Table S1).

The shallow and deeper populations sampled in the Atlantic and Mediterranean do not coincide with the same exact depths, as vertical distribution limits varied between sites. Thus, comparisons of genetic diversity levels between shallow and deeper populations were made at the site scale, and not between sites. Allelic richness, number of private alleles and gene diversity (expected heterozygosity) per population (shallow and deeper) were standardized to the smallest sample size found within sites using $10^4$ randomizations. Significant differences in mean diversity levels within each site were tested using a non-parametric Wilcoxon signed-rank test with $10^4$ randomizations (*Assis et al., 2018*).

To infer the drivers shaping genetic variation among sites, genetic diversity estimates were standardized to the smallest size of all samples using $10^4$ randomizations. The estimates of diversity were modelled with linear regression against depth (null model) and a set of important environmental predictors affecting the physiology of *P. clavata* (Maximum bottom temperature; *Boavida et al., 2016b*) and proxies of essential resources (minimum bottom productivity, minimum bottom current velocity and slope, for food intake; (*Boavida et al., 2016b*). Other important predictors were not considered since (1) they did

not vary between sampling sites (e.g., salinity) or (2) were correlated with productivity (e.g., inorganic nutrients such as nitrates and phosphates). The predictors were developed with three-dimensional profiles of monthly data compiled from the Global Ocean Physics Reanalysis (ORAP; http://www.marine.copernicus.eu/) and the Biogeochemistry Non-Assimilative Hindcast Simulation (PISCES; http://www.marine.copernicus.eu/). Bottom environmental data for each population was obtained using trilinear interpolation (*Assis et al., 2017*) weighting location (longitude and latitude) and depth. Long-term minimum (productivity and currents) and maximum (temperature) extremes were averaged for the years 2000–2014. Slope was computed using the "terrain" function of the R package "raster" (3.5.2 version; *R Core Team, 2018*) in bathymetry.

Given the sampling design (no precise coincidence of depths between sites) we adopted the Generalized Linear Mixed Models (GLMM; *Bolker et al., 2009*) framework. This is suitable for modelling unbalanced designs (*Zhang & Chen, 2013*) and complex spatial and temporal correlation structures since it accounts for dependencies within hierarchical groups by introducing random effects (*Bolker et al., 2009*). While these effects generally comprise the blocking in experimental treatments, they can be used to encompass variation among geographical regions (*Dormann et al., 2007*). Accordingly, "site" was included in the models as a random effect term (*Zuur, Ieno & Smith, 2007*; *Ludwig et al., 2012*). The best-fit selection between the null and the environmental models followed the relative Akaike Information Criterion (AIC), while goodness-of-fit was inferred with the R2GLMM algorithm from *Edwards et al. (2008)*. Partial dependence plots were also produced to illustrate the effect of predictors on the response of models, by accounting for the mean effect of all other predictors (*Elith, Leathwick & Hastie, 2008*). All models were performed in the R environment (3.5.2 version; *R Core Team, 2018*).

To infer larval dispersal potential, pairwise genetic differentiation was estimated between and within sampled populations (i.e., shallow vs. deeper populations) using $F_{ST}$ estimator and an analysis of molecular variance (AMOVA) based on allele frequencies, computed with Genodive (*Meirmans & Van Tienderen, 2004*) under $10^4$ randomizations (*Assis et al., 2013*).

Evidence of genetic bottlenecks were inferred by testing for heterozygosity excess (*Piry, Luikart & Cornuet, 1999*). This is rooted on the assumption that populations that have recently experienced a bottleneck event are predicted to temporarily reduce allelic diversity at a faster rate than heterozygosity. Such an excess in heterozygosity rate was tested for each population with the software Bottleneck (*Piry, Luikart & Cornuet, 1999*) using $10^4$ simulations. We used the suggested and more appropriate (realistic) parameters for microsatellites: the Two-Phase Model (TPM) (*Luikart & Cornuet, 1998*; *Piry, Luikart & Cornuet, 1999*) with a step in mutations of 0.9 (ps) and a variance in mutations of 12 (*Piry, Luikart & Cornuet, 1999*; *Busch, Waser & DeWoody, 2007*). Because the genetic dataset has less than 20 loci, we used the Wilcoxon test to address the null hypothesis of no heterozygosity excess (on average) across loci (*Cornuet & Luikart, 1996*; *Luikart & Cornuet, 1998*).

## RESULTS

Genetic diversity as allelic richness (A) and number of private alleles (PA) was consistently higher in deeper populations, with the exception of only one site (Petit Congloue; Figs. 2A,

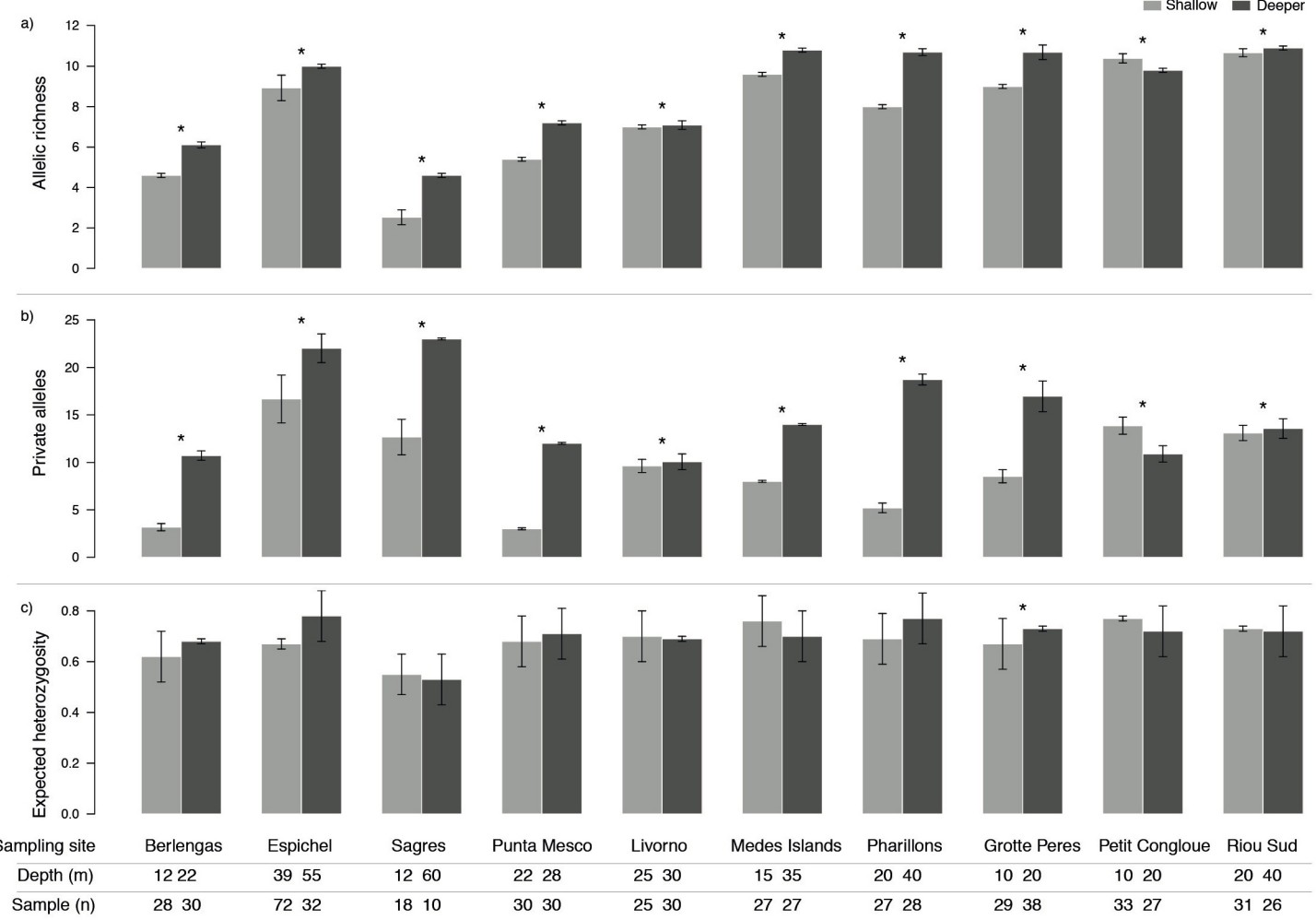

**Figure 2** **Genetic diversity (mean ± SD) as (A) allelic richness, (B) private alleles and (C) expected heterozygosity of *Paramuricea clavata* in shallow (light grey) and deeper (dark grey) populations.** Asterisks indicate significant differences in diversity levels ($P < 0.05$). Sampling site name, depth ($m$) and number of samples ($n$) are described for each population.

2B). The greatest difference in A and PA between shallow and deeper populations was observed in Pharillions and Sagres (A: 2.7 and 2.07; PA: 13.52 and 10.33, respectively), while the smallest was observed in Livorno (A: 0.09 and PA: 0.43). The variation in expected heterozygosity (He) did not follow a clear pattern, with only one sampling site (Grotte Peres) having a higher He in its deeper population (Fig. 2C).

The null linear regression models using depth alone were outperformed by the models using environmental predictors (environmental models showed lower AIC and higher $R^2$; Table 1). Both approaches found significant relationships while fitting the predictors against standardized allelic richness and number of private alleles (Table 1), however, they failed to explain the variability in expected heterozygosity. The environmental models showing better fitting were mostly explained by temperature alone (Table 1). This predictor

**Table 1 Summary of linear regression models testing genetic diversity indices against depth and a set of environmental predictors.** Akaike Information Criterion, $R$-squared and $p$-value scores are shown for each model. Bold represents higher values when comparisons were made.

| | | Allelic richness | | | Private alleles | | | Expected heterozygosity | | |
|---|---|---|---|---|---|---|---|---|---|---|
| Model type | Predictors | AIC | $R^2$ | $p$-value | AIC | $R^2$ | $p$-value | AIC | $R^2$ | $p$-value |
| Null | Depth | 71.05 | 0.22 | 0.01 | 92.84 | 0.60 | 0.01 | −35.26 | 0.01 | 0.77 |
| Environmental | Model effect | **46.83** | **0.63** | | **74.31** | **0.62** | | **−39.33** | **0.43** | |
| | Min. current velocity | | | 0.98 | | | 0.45 | | | 0.29 |
| | Min. productivity | | | 0.99 | | | 0.95 | | | 0.81 |
| | Slope | | | 0.72 | | | 0.56 | | | 0.13 |
| | Max. temperature | | | 0.01 | | | 0.01 | | | 0.53 |

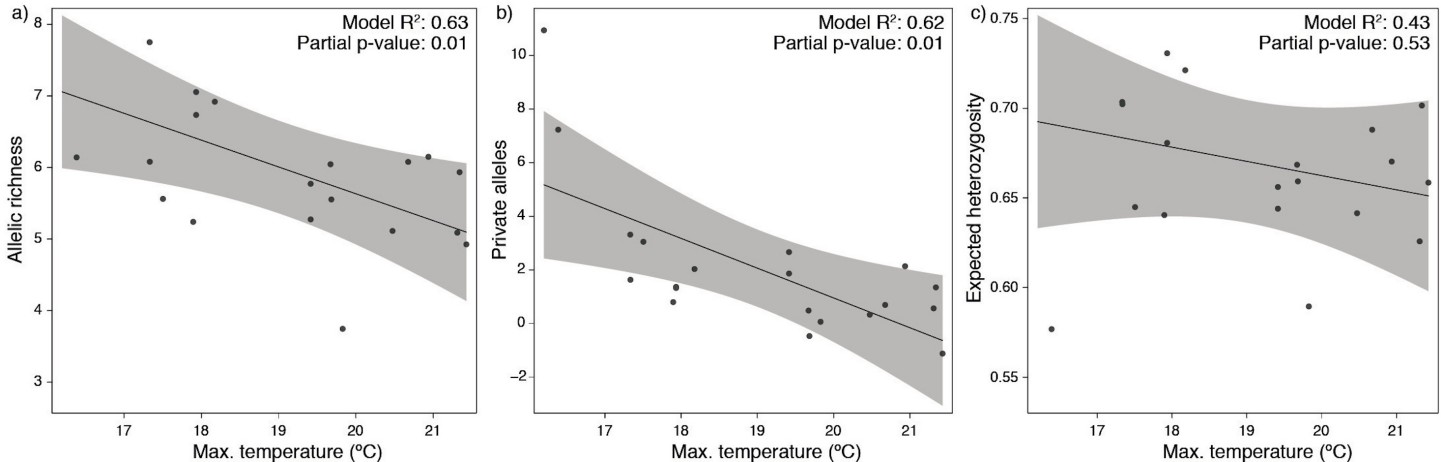

**Figure 3 Partial dependence functions depicting the effect of bottom temperature on (A) allelic richness, (B) private alleles and (C) expected heterozygosity.**

produced a negative response on the models, with lower allelic richness and private alleles with increasing bottom temperatures (Fig. 3).

Genetic differentiation ($F_{ST}$) was higher between than within sampling sites, with two exceptions for Sagres and Riou Sud (Fig. 4). The former site (i.e., Sagres) showed the highest differentiation of all pairwise comparisons at the site scale (i.e., shallow vs. deeper sample; $F_{ST} \approx 0.4$; Table S2). No significant differentiation was found within 7 Mediterranean sites (Punta Mesco, Livorno, Pota Del Llop, Grotte Peres and Petit Congloue; Table S2) and between the two populations of Grotte Peres and the deeper population of Petit Congloue (Table S2), which are approximately 2 km far apart.

Bottleneck events (heterozygosity excess) were detected in the shallow populations of Berlengas and Punta Mesco (Table 1; S3). All remaining populations showed no heterozygosity excess (on average) across loci (Table 1; S3).
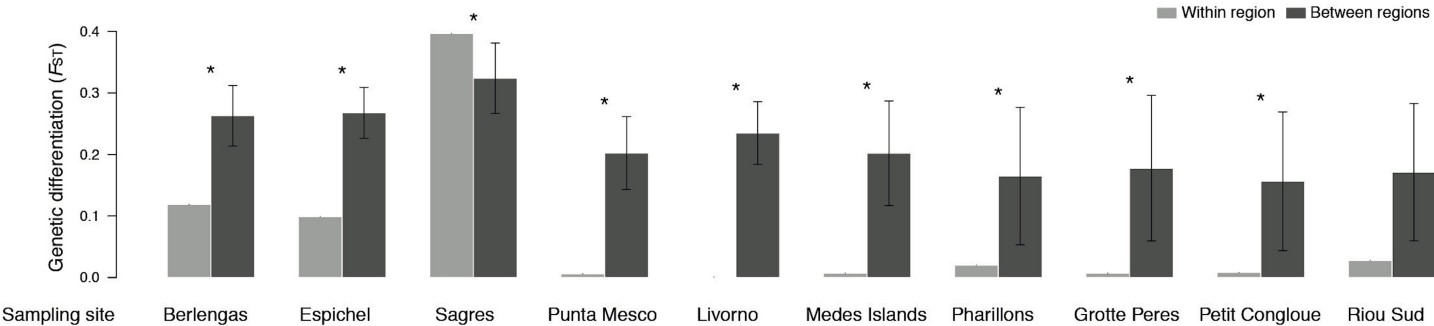

**Figure 4** Pairwise genetic differentiation $F_{ST}$ (mean ± SD) within sampling site (i.e., shallow vs. deeper populations; light grey) and between sampling sites (dark grey). Asterisks indicate significant differences in genetic differentiation levels ($P < 0.05$).

## DISCUSSION

The distribution of genetic diversity corroborated the expectations associating extreme environmental changes with the reduction of effective population sizes and genetic diversity levels (*Garrabou et al., 2009*). Our model species *P. clavata* consistently showed less allelic richness, a smaller number of private alleles and bottleneck signs (for two sites) in shallow waters populations. This pattern found across the Atlantic Ocean and Mediterranean Sea was explained by the negative relationship between thermal stress and depth, and not by the used proxies of essential resources (i.e., productivity, current velocity and slope). In fact, the examination of daily time series of bottom temperatures for each site and depth range (S4) shows shallow water masses recurrently surpassing the physiological threshold of 24–25 °C from which mortality occurs (*Previati et al., 2010*; *Pairaud et al., 2014*; *Boavida et al., 2016b*), particularly in the Mediterranean Sea. This is in line with previous studies linking mass-mortality events of *P. clavata* with marine heatwaves, which reported increased mortality in more shallow environments (*Cerrano et al., 2005*; *Linares et al., 2005*). Still, the observed differences in genetic diversity may have resulted from additional factors beyond temperature changes, not considered in our study. For instance, the scarcity of favourable rocky bottoms may strongly reduce population sizes of *P. clavata* (*Linares et al., 2008*; *Boavida et al., 2016b*). This could be a plausible explanation for the one exception found in the diversity levels of deep populations. The lower diversity in the deeper population of Petite Congloue may result from a smaller and patchier distribution of individuals, if the availability of rocky bottoms is limited. In a particular survey, Gori et al., 2011 found the density of *P. clavata* decreasing with depth, as populations were highly dependent on the presence of rocky vertical walls in shallower waters. In our study, the availability of rocky bottoms was not considered, although slope may be considered as a proxy for such environmental factor at biogeographical scales (e.g., *Boavida et al., 2016b*), it might not have reflected site scale habitat changes between our sampling depths. Further, competition with other species, such as photosynthetic algae (*Zabala & Ballesteros, 1989*), together with higher hydrodynamics regimes and human induced disturbances (e.g., anchoring, scuba diving), may further help explaining the overall pattern found, restricting
large, continuous and richer populations to deeper sites (*Harmelin & Marinopoulos, 1994*; *Cúrdia et al., 2013*).

Contrarily to allelic richness and number of private alleles, gene diversity (i.e., expected heterozygosity) did not correlated with depth, nor with the additional predictors considered in regression analyses. Gene diversity is not as sensitive as allelic richness to detecting historical population changes (*Leberg, 1992*; *Petit, El Mousadik & Pons, 1998*; *Spencer, Neigel & Leberg, 2000*) since drift resulting from population size reductions theoretically affects more the rare alleles than the frequent ones (*Nei, Maruyama & Chakraborty, 1975*). Accordingly, the diversity pattern found in deeper populations implies persistence without significant population reduction, raising the hypothesis of deep refugia for *P. clavata*. These cryptic populations may play an important role in buffering the loss of diversity in shallow waters, as reported for other corals (*Bongaerts et al., 2010*; *Bongaerts et al., 2017*; *Smith et al., 2014*). Similar to terrestrial elevational refugia, the stability of deep environments may be an important mechanism safeguarding regional genetic diversity for the species as a whole (*Epps et al., 2006*; *Assis et al., 2016*; *Lourenço et al., 2016*).

While diversity levels of *P. clavata* followed the patterns of thermal stress, the bottleneck tests did not provide a strong empirical support of recent bottlenecks in shallow populations (exceptions for Berlengas e Punta Mesco). Although recent environmental changes may account for genetic diversity losses, the fluctuations of past climate changes may further shape the genetic structure of marine species, both latitudinally and along depth gradients (*Maggs et al., 2008*; *Provan, 2013*; *Assis et al., 2014*; *Assis et al., 2016*; *Assis et al., 2018*; *Neiva et al., 2016*). The lower diversity found in shallow populations could have resulted from past climate extremes, an historical effect that cannot be detected with tests based on heterozygosity excess (*Peery et al., 2012*). Also, one cannot discard the limited statistical power of these tests when using small sample sizes, as in our case (small number of individuals and loci; (*Piry, Luikart & Cornuet, 1999*; *Peery et al., 2012*). Our approach could have overlooked actual population declines. For instance, previous studies on *P. clavata*, with similar sample sizes, drew unambiguous conclusions by failing to detect bottlenecks in populations that actually suffered mass-mortality events (*Pilczynska et al., 2016*; *Padrón et al., 2018*).

Population differentiation was found higher between than within sampled sites. This is in line with the low dispersal potential of the species, posing highly structured gene pools throughout the Atlantic Ocean and the Mediterranean Sea (*Linares et al., 2007*; *Mokhtar-Jamaï et al., 2011*; *Pilczynska et al., 2016*; *Padrón et al., 2018*). At the site scale, the non-significant differentiation levels (Punta Mesco, Livorno, Pota Del Llop, Grotte Peres and Petit Congloue) suggest that gene flow and admixture occurs along depth ranges. While summer thermoclines were thought to be a major force structuring genetic diversity of gorgonian species (*Costantini et al., 2011*; *Costantini et al., 2016*), our results do not support such hypothesis. The shallow populations of *P. clavata* harbouring lower diversity levels are not in complete isolation from those in deep richer areas, as previously suggested for the species (*Cerrano & Bavestrello, 2008*). However, contrasting patterns of coral gene flow can occur across different environments (e.g., *Van Oppen et al., 2011*; *Mokhtar-Jamaï et al., 2011*; *Pilczynska et al., 2016*). For instance, the higher within site differentiation in

Sagres can result from the alongshore coastal circulation regime in the Gulf of Cadiz that extends to the Sagres region (*Garel et al., 2016*), promoting high connectivity levels along deep neighbouring populations, or from the fact that the shallow population of Sagres was located within a cave, possibly, further hindering connectivity. This was an exception though, as the overall connectivity pattern found is one of restricted connectivity between deeper populations. Thus, their diversity levels may have resulted from persistence alone, and not by admixture processes, as also suggested by the number of private alleles. The differentiation pattern further allows concluding that connectivity along depth ranges (within site) does not seem to completely offset the more detrimental effects occurring in shallow water populations, reducing genetic diversity of *P. clavata*, as observed.

Our findings of deep isolated richer populations have significant conservation value for the species as a whole. Future environmental changes are predicted to produce a major redistribution of marine biodiversity in the Atlantic Ocean and the Mediterranean Sea (*Albouy et al., 2013*; *Assis, Araújo & Serrão, 2017*), particularly in the business as usual climate scenario (RCP8.5; *Assis, Araújo & Serrão, 2017*; *Assis et al., 2017*). The reduction of biodiversity through the loss of rich and distinct gene pools found in deep populations may contribute to the loss of genes responsible for the species adaptation and evolution (*Hughes & Stachowicz, 2004*; *Hampe & Petit, 2005*; *Reusch et al., 2005*). The identified genetically rich populations of *P. clavata* represent an important baseline for Climate Change Integrated Conservation Strategies for phylogeographic lineages (*Hannah, Midgley & Millar, 2002*).

## CONCLUSIONS

This study demonstrates that deeper marine populations can consistently harbour higher genetic diversity than those in shallower environments. Theoretical regression modelling using environmental data showed that such genetic pattern may arise from thermal stress alone, reducing population sizes and diversity levels. These findings raised the hypothesis of deep refugia for *P. clavata*, with deeper populations in more stable environments safeguarding the species biodiversity across the Mediterranean and Atlantic populations. In fact, the analyses of temperature time-series showed shallow populations exposed to temperatures surpassing the species thermal tolerance, often for as long as over 2 months, while deep populations were almost never exposed to such conditions. The patterns of population differentiation revealed that gene flow occurs between shallow and deeper populations, although not in sufficient levels to homogenise depth differences in diversity levels. Empirical evidence is provided for deep persistent populations of gorgonians with the potential to safeguard richer and unique gene pools. While additional genetic data could better support the major findings, particularly by increasing the power of bottleneck tests (more molecular markers and sampling sites), this study represents a timely baseline to conserve populations with higher conservation value in risk of disappearing.

### Funding

This study was supported by financial support to CESAM (UID/AMB/50017/2019), to FCT/MCTES through national funds, and the co-funding by the FEDER, within the PT2020 Partnership Agreement and Compete 2020. This study was also supported by a Pew Marine Fellowship (USA), the National Geographic Channel through project Deep Reefs, a National Geographic/Waitt grant (no. W153-11), the InAqua Conservation Fund (Oceanário de Lisboa), the European Regional Development Fund (COMPETE program) and the Foundation for Science and Technology (FCT) of Portugal through postdoctoral fellowship SFRH/BPD/111003/2015 and programs CCMAR/Multi/04326/2013, UID/MAR/04292/2013 and PTDC/BIA-BIC/114526/2009 (DiverseShores—Testing associations between genetic and community diversity in European rocky shore environments). Joanna Pilczynska was supported by a MARES Grant. MARES is a Joint Doctorate programme selected under Erasmus Mundus coordinated by Ghent University (FPA 2011-0016). The funders had no role in study design, data collection and analysis, decision to publish, or preparation of the manuscript.

### Grant Disclosures

The following grant information was disclosed by the authors:
CESAM: UID/AMB/50017/2019.
FCT/MCTES through national funds.
FEDER.
Pew Marine Fellowship (USA), the National Geographic Channel.
National Geographic/Waitt: W153-11.
InAqua Conservation Fund (Oceanário de Lisboa).
European Regional Development Fund (COMPETE program).
Foundation for Science and Technology (FCT) of Portugal through postdoctoral fellowship: SFRH/BPD/111003/2015, CCMAR/Multi/04326/2013, UID/MAR/04292/2013, PTDC/BIA-BIC/114526/2009.
DiverseShores—Testing associations.
Erasmus Mundus coordinated by Ghent University: FPA 2011-0016.

### Competing Interests

The authors declare there are no competing interests.

### Author Contributions

- Joanna Pilczynska and Joana Boavida conceived and designed the experiments, performed the experiments, analyzed the data, contributed reagents/materials/analysis tools, prepared figures and/or tables, authored or reviewed drafts of the paper, approved the final draft.
- Silvia Cocito, Ester A Serrão and Henrique Queiroga conceived and designed the experiments, analyzed the data, contributed reagents/materials/analysis tools, authored or reviewed drafts of the paper, approved the final draft.

- Jorge Assis performed the experiments, analyzed the data, contributed reagents/materials/analysis tools, prepared figures and/or tables, authored or reviewed drafts of the paper, approved the final draft.
- Eliza Fragkopoulou analyzed the data, contributed reagents/materials/analysis tools, prepared figures and/or tables, authored or reviewed drafts of the paper, approved the final draft.

## Field Study Permissions

The following information was supplied relating to field study approvals (i.e., approving body and any reference numbers):

1. Parco Nazionale delle Cinque Terre Area Marina Protetta delle Cinque Terre.
2. Instituto de Concervação da Natureza e da Biodiversidade

## Data Availability

Microsatellite data for Paramuricea clavata samples are provided in Dataset S1. The data are available in GEN format and can be used in GENEPOP software package (http://genepop.curtin.edu.au). Each sample is named after the initials of the sampling location, where the last letter "S" stands for shallow populations and "D" stands for deep populations.

## Supplemental Information

Supplemental information for this article can be found online at http://dx.doi.org/10.7717/peerj.6794#supplemental-information.

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
