# Peer review of "Genetic diversity increases with depth in red gorgonian populations of the Mediterranean Sea and the Atlantic Ocean"

_PeerJ, doi:10.7717/peerj.6794_

## Round 0.1 · original submission · Major Revisions

· Academic Editor

Major Revisions

The reviewers were enthusiastic about the potential contributions of the study, however, most of the examiners identified several aspects of the manuscript that require major revisions before this can be considered for publication. In general, there is a considerable lack of literature review of previous work. Furthermore, the discussion needs to be revised, as some of the interpretations were considered speculative and not well supported from the actual data

Also, suggestions to improve the clarity of figures have been proposed. More methodological information has been requested to be included in the revised manuscript. Make sure you address all the comments provided by the reviewers.

Reviewer 1 ·

Basic reporting

There is insufficient methodology in this submission to allow others to evaluate the study without consulting another study (Pilczynska et al. 2016) which is unacceptable. Include sufficient methods to be self-contained.

Experimental design

1. The hypotheses need to be better formulated (remove “might” at the very least). I would seriously think about the fact that the populations have maximum depth limits too so the populations at their deeper limit will suffer selective pressure too from some other factor –the Mediterranean populations were between 20-30m so I assume that collections were made near maximum depth limits for the species there. It is possible that maximum population stability is at some intermediate depth.
2. I can’t assess the collection method used but I don’t think a linear regression is an appropriate test for the hypotheses (“We hypothesized that diversity was reduced in shallow populations due to higher environmental variability”).
3. The authors state that depth ranges differed between Atlantic and Mediterranean populations but then lump them together for the analyses. They should have collected deep and shallow individuals at each site and compared them so as to avoid confounding space and depth.

Validity of the findings

1. It is not clear to me what is the difference between the data in Figure 2 and Table 1 – aren’t they derived from the same dataset? Why are the scales different? What are the +/- in the Table? Are the data in Fig 2 means – if so why not include a measure of variation?

2. The ms focuses on temperature as the selective pressure as they can model it but other environmental factors may be selective and may also operate over a depth range too; pressure, density, salinity, other species, etc. So it is difficult to accept that temperature is THE factor in a correlation study in which temperature was not actually measured.

3. In order to be convincing that there is a clear pattern the reader would need to see more data, and from deeper and more points. At the moment the deepest sites (Lagos and Sacres 2, 60 m) actually have the lowest diversity and are amongst the least rich in alleles.

Reviewer 2 ·

Basic reporting

The manuscript is clear and concise, novel for the species analysed and also for the findings. Literature lackes of several crucial references that would be essential to make a clearer interpretation of the data set. The comparison of the two areas is very interesting, but a more in-depth interpretation of the physical factors as drivers of the potential diferences in genetic diversity can be made. The future readers will find useful having a more accurate description of what may be the environmental constrains in the Atlantic and Mediterranean sea for the larval dispersion. Figures and tables are explicit and well done. See the pdf for further comments.

Experimental design

The experimental design is quite well conceived. However the real constraint is that both populations are a little bit difficult to compare, due to the fact that they are sampled at different depths in the twilight zone. This fact is not, however, a definitive problem for the manuscript. Genetic analysis are on line with previous works (and I congratulate the authors, Paramuricea clavata is difficult to analyse). See more comments on the pdf.

Validity of the findings

The original question has to be clarified. I find that the genetic diversity that the authors claim is increasing with depth in Paramuricea clavata has to be better explained. See more comments in the pdf.

Additional comments

I have one general comment for the authors that is really important: you do not take into account the reprodutive traits and the larval behaviour of the species. And this is, simply, essential. There are many works explaining reproductive cycle, effort, differences among populations, first larval stages, etc. It is supposed that your work is about connectivity, after all. And no relation between your findings and the potential larval dispersion (and the time of the year in which this happens) is written. I suggest to implement this information to enrico your conclusions (see pdf).

Annotated reviews are not available for download in order to protect the identity of reviewers who chose to remain anonymous.

Reviewer 3 ·

Basic reporting

The manuscript is written in a good english but present a lack in literature references and background about the previous works made on the same species and on the "refugia hypothesis".
The figures are not clear and need a revision.
Based on the hypothesis that Authors want to test, there is no a specific sampling design and based on their results, the discussion is to much speculative.
See my detailed comments on the general comments session.

Experimental design

see my comment on the general comments section.

Validity of the findings

see my comment on the general comments section.

Additional comments

Dear Editor,
The manuscript “Genetic diversity increases with depth in red gorgonian populations of the Mediterranean Sea and the Atlantic Ocean” by Pilczynska and colleagues investigate the change of genetic variability from shallow to deep populations of the red gorgonian Paramuricea clavata sampled both in the Mediterranean Sea and Atlantic Ocean. The main result of their study is that genetic variability varies with depth, but not that there is an increase in genetic variability with depth. One of the main concerns regarding this study is the choice of the sampled populations. Authors sampled populations at different depths in the Atlantic and at different depths in the Mediterranean Sea. If they want to test the effect of the depth on the genetic variability they should samples populations at the same depth in both basins or along a bathymetric gradient. Moreover, they should sample populations in the same area at two different depths to not be biased by a geographical sampling. They, in fact, use temperature time-series as a proxy for the depth effect but in my opinion, other parameters can play a role in the genetic variability pattern. Then, they cannot disentangle the effect of genetic connectivity in determining the patter of genetic variability because some populations could show less genetic variability because are more isolated than other. Finally, I think that some important references are missed in the manuscript because there are several studies that have studied the effect of the depth on Mediterranean gorgonian populations (e.g. Cánovas-Molina et al. (2018); Padron et al. 2018; Costantini et al. 2016; Arizmendi-Mejía et al. 2015; Costantini et al. 2011; Mokhtar‐Jamaï et al. 2011). All these information should be taken into account in the new version of the manuscript. Regarding the sampling design, I found it not really exhaustive. Based on the aim of the paper, in my opinion, Authors should have performed a sampling design specifically addressed to answer their question. Based on their sampling design other causes not related to the depth should explain the genetic variability pattern observed. Maybe a more exhaustive description of the sampled sites will implement the paper. Moreover, the Authors sampled a lot of colonies but at the end perform all results using only ten of them. Finally, they have to explain why they analyse only 5 of the 10 available species-specific microsatellite loci. Finally, Authors published in 2016 a similar work using some of the same populations analysed here and it is strange that they did not discuss ever the new results in function of the old ones. The discussion, in my opinion needs a complete revision. In fact, as said before, some important references are missed and a more detailed discussion of the high variability observed within basin and across depth should be taken into account. To me they cannot conclude that deep population of Paramuricea clavata represent refugia for shallow water populations (see for example the reduction of genetic variability at 60 m depth).

ABSTRACT:
Line 29: Delete “along the first tens of meters”. To me it is too precise, is not always like this. It depends on species and habitat and geographic zone.
Lines 32-34: is not completely clear. Rewrite this sentence.
Lines 39-40: “genetic diversity IN shallow and deep populations”

INTRODUCTION:
Line 63: add “of the population”
Line 66-68. Thermal stress is only one example of processes that can influence selection. Try to rephrase this sentence.
Line 68: “long time” instead than “long-term”
Line 80: It is not completely true that the deep refugia hypothesis is rarely tested. See for example (only to cite some of them):
Bongaerts, P., Riginos, C., Brunner, R., Englebert, N., Smith, S. R., & Hoegh-Guldberg, O. (2017). Deep reefs are not universal refuges: reseeding potential varies among coral species. Science Advances, 3(2), e1602373.
Bongaerts, P., Ridgway, T., Sampayo, E. M., & Hoegh-Guldberg, O. (2010). Assessing the ‘deep reef refugia’hypothesis: focus on Caribbean reefs. Coral reefs, 29(2), 309-327.
Van Oppen, M. J., Bongaerts, P. I. M., Underwood, J. N., Peplow, L. M., & Cooper, T. F. (2011). The role of deep reefs in shallow reef recovery: an assessment of vertical connectivity in a brooding coral from west and east Australia. Molecular ecology, 20(8), 1647-1660.
Moreover there are also studies made on Mediterranean and on gorgonians that evaluate the effect on the depth on genetic variability of the species that should be taken in account.
Line 94: Eliminate “and might…” It is too speculative without references.
Line 96: P. clavata populations
Line 97: I think that Authors have to explain why the compared Atlantic vs. Mediterranean populations. Do they expect different or similar results? Based on what assumptions? I think that they have to clarify better. Moreover, I think that we cannot disentangle genetic variability from genetic connectivity because maybe populations in the deep show high genetic variability because are more connected among them or vice versa.

METHODS:
Line 108: What Authors mean with “environmental gradient”? They mean “ temperature” or also other abiotic parameters? I think that they have to explicit better this part because is the more critical (see comment above). Moreover, is not completely true that in the Mediterranean the vertical distribution of Paramuricea clavata is within the first 30 meters. There are some populations below this depth (e.g. Spain, Gori et al. 2011; Ligurian coast, Padron et al. 2018; below 50 meters in the Calabria coasts (Bo et al. 2009)). In fact, the bathymetric distribution range of the species in the Mediterranean Sea goes from around 15 meters to more than 200 meters.
Line 109: How many colonies per site were sampled? Please specify.
Lines 112-113: what kind of “other sampling methods”? Please, explain. Based on the aim of the paper I think that more information regarding the sampling design should be given here.
Line 126-127: This analysis is not clear. What do Authors mean for “regional mean”?
Line 129: What kind of software the Authors have used to determine genetic variability parameters? And for the linear regression model?

RESULTS:
Line 150: Should be interesting to test the differences in terms of genetic variability between basins and between shallow and depth sites within basin. “Generally” does mean nothing from a scientific point of view; such as “nearly”.
Line 151: correct “max PA less than 4”. Max PA should be a specific value.
Line 151: Do Authors consider Espichel1 shallow o deep? Authors have to specify that here they talk about Espichel2 (based on what Author have written, 50-meter depth is considered “shallow water”). Moreover, also Espichel2 show a high value of genetic variability compared to the deeper populations (Sagre2 and Lagos). In the Material, Authors stated that deep populations are those found at 60-meter depth. I think that in the MM session Authors have to better explain what populations they consider shallow and depth and why.
Lines 152: What about the only other two deep populations (Sagres 2 and Lagos)? They do not show high values of genetic variability.
Lines 153-154: Yes, there is a significant correlation with depth in A and PA but still deeper populations showed low values compared to population found at 50-meter depth. Authors should describe better the correlation patterns.
Lines 157: 12°C
Lines 157: I think that this is the main result: “genetic diversity vary with depth”. Not that there is an increase of genetic variability with depth.

DISCUSSION

Lines 169: It is strange that Authors here did not use the existing references on gorgonians regarding the possible bottleneck. In fact, for example, a paper carried out by the same Authors (Pilczynska et al. 2016) does not show evidence of a recent bottleneck in the same shallow populations analysed here. Conversely, Padron et al. (2018) evidence bottleneck in some Ligurian populations at a deeper depth. I think that all this information should be discussed and taken into account in the discussion.
Line 171: Please, specify what is the “thermal threshold”.
Lines 174: Authors should explain that Boavida et al. is based on an ecological niche modelling. There are other references that should be taken in account here based on other papers carried out in particular on the northwestern Mediterranean Sea (e.g. Crisci et al. 2011).

Lines 178: The “refugia hypothesis” should be more detailed. More important references are missing here depending the contrasting results of this hypothesis and how it varies depending on the geographical areas and species.
Line 185: Why Authors did not test the presence of bottleneck? This is an analysis that should be added in the manuscript to confirm or not they results.
Lines 187-197: All this part should be rewritten. Authors should be discussing more in detail their results, explaining the differences that they observed among sites and among depth. They results do not allow to evidence a so clear common pattern of variability.
Line 203: It is the first time that the “connectivity” is taken in account in the manuscript. I think that this should be also introduced before.
Lines 216: Again, I don’t think that they can generalize here. They did not test other invertebrates. Maybe other species show a completely different pattern (see for example other gorgonians).
Lines 222: the paper is focus on temperature variability not “environmental” in general.
Lines 226: “higher and more variable”
Lines 227: as said before, the bottleneck hypothesis should be tested here.


Figure 3: This figure should be redesigned. Is not clear. Some Atlantic populations are missing (e.g. Espichel, Lagos). Moreover and indication of the depth where the temperature was measured is needed. Again, to me, shallow and deep in the context of the manuscript need to be better explicated.

---

## Round 0.2 · Minor Revisions

· Academic Editor

Minor Revisions

I am a Section Editor on the journal and I have been asked to make the decision on your revision as the original Editor is out of contact and not able to do so.

Thank you for the detailed revision in response to referee comments. Both referees agree that the manuscript is greatly improved from the first submission but still requires minor revisions before it is likely to become acceptable for publication.

The first issue is easily addressed - as reviewer #1 points out, there are a variety of usage and grammar issues that have crept into the revision, and the manuscript needs a careful revision to ensure proper English usage throughout. Likewise with correcting the captions on your figures.
The second issue pointed out by the same referee is that your analysis does not include slope as a factor. The referee points out that the Boavida et al. (2016) study finds slope as the primary factor, which coincides with our findings here in Hawaii, where slope was one of the major predictors of mesophotic coral community structure (Veazey et al. 2016, PeerJ 4:e2189). I, like the referee, would have liked to see slope as an explicit factor in your analyses. I also take their point that the choice of variables stacks the deck to favor temperature as the primary factor - you mention unmeasured variables may play a role in the discussion, but still draw strong conclusions about the role of temperature on these populations. I believe that a slightly more balanced discussion of such factors would elicit less resistance from referees and future readers alike.

I understand that it is not always possible to redo analyses, but given these previous findings together with the intuition of the referee as well as my own, it seems that at least some explicit discussion of slope in the manuscript is warranted.

I look forward to your revised manuscript.

Reviewer 1 ·

Basic reporting

Some English mistakes have crept in throughout the new version which needs to be revised (ex. L255-6 “AS a whole”; L253 “diversity IN shallow waters”; L257-8 “the tests aiming AT size reduction” (by the way, reduction of the size of what?); etc.)
The captions of figures 3 and 4 are swapped.

Experimental design

My main criticism of this new approach is that the chosen environmental variables kind of stack the deck against finding differences in other variables and in favor of temperature. I understand the rationale for using maximum temperature as a variable. I imagine that minimum current velocity is less variable among sites and with depth than mean or maximum current velocity; and I don’t really see what physiological basis is behind this choice; minimum productivity is the same case. The authors justify their choice by citing Boavida et al (2016) who actually identify slope as the most important factor in their modelling study, followed by temperature minima and maxima, then silicate maxima. Slope was not used here. The question is then: can slope be systematically related to depth? – if so it could be confunding. I understand that it is not possible to include all variables and the study of Boavida et al (2016) points to the direction to be taken, but their main factor is ignored. The authors seem to agree (L233-240), so why not include slope in the study?

Validity of the findings

See comment above.

Additional comments

The authors should be congratulated in taking on most of the criticisms of the first draft and this paper has been substantially modified from the first version, for the better.

Reviewer 2 ·

Basic reporting

The authors made substantial improvements on the manuscript. In fact, they correctly addressed, from my point of view, two of the most problematic questions posed by me:

"Both populations are a little bit difficult to compare, due to the fact that they are sampled at different depths in the twilight zone." is now clearer; the changed text is satisfactory

"No one word about the reproductive and dispersal patterns of the species." is now substantially changed

Experimental design

They made a good job readdressing the questions made not only by myself but also by the other two referees. The Experimental design is now clearer.

Validity of the findings

I had no doubt in the previous review that the findings were relevant and valid.

Additional comments

From my side, the paper has the improvement needed to be published. The "original question" has been clarified and the robustness of the paper is now improved, with a concise target.

---

## Round 0.3 · accepted · Accept

· Academic Editor

Accept

Thank you for your revisions to the manuscript and for adding slope as an explicit factor to the analysis. I assumed that you likely had the comparison already available given previous work of some of the authors on this manuscript, so I am glad to see your response. The correction of typos and grammar mistakes together with the addition of slope to the analyses addresses the concerns of the referee without altering any of your original conclusions. In my opinion, the additions you have made to the manuscript have improved it to the point where I am happy to accept your submission and move it forward into production.

#